# Processing Effects on the Through-Plane Electrical Conductivities and Tensile Strengths of Microcellular-Injection-Molded Polypropylene Composites with Carbon Fibers

**DOI:** 10.3390/polym14163251

**Published:** 2022-08-10

**Authors:** Shia-Chung Chen, Ming-Yuan Jien, Chi-Chuan Hsu, Shyh-Shin Hwang, Ching-Te Feng

**Affiliations:** 1R&D Center for Smart Manufacturing, Chung Yuan Christian University, Taoyuan 32023, Taiwan; 2R&D Center for Semiconductor Carrier, Chung Yuan Christian University, Taoyuan 32023, Taiwan; 3Department of Mechanical Engineering, Chung Yuan Christian University, Taoyuan 32023, Taiwan; 4Department of Mechanical Engineering, Chien-Hsin University of Science and Technology, Taoyuan 32031, Taiwan

**Keywords:** fiber orientation, microcellular foaming, gas counter pressure, through-plane electrical conductivity, tensile strength, polypropylene composites

## Abstract

Polymers reinforced with conducting fibers to achieve electrical conductivity have attracted remarkable attention in several engineering applications, and injection molding provides a cost-effective way for mass production. However, the electrical performance usually varies with the molding conditions. Moreover, high added content of conducting fibers usually results in molding difficulties. In this study, we propose using microcellular (MuCell) injection molding for polypropylene (PP)/carbon fiber (CF, 20, and 30 wt%) composites and hope that the MuCell injection molding process can improve both electrical and mechanical performance as compared with conventional injection molded (CIM) parts under the same CF content. Both molding techniques were also employed with and without gas counter pressure (GCP), and the overall fiber orientation, through-plane electrical conductivity (TPEC), and tensile strength (TS) of the composites were characterized. Based on the various processing technologies, the results can be described in four aspects: (1) Compared with CIM, microcellular foaming significantly influenced the fiber orientation, and the TPECs of the samples with 20 and 30 wt% CF were 18–78 and 5–8 times higher than those of the corresponding samples molded by CIM, respectively; (2) when GCP was employed in the CIM process, the TPEC of the samples with 20 and 30 wt% CF increased by 3 and 2 times, respectively. Similar results were obtained in the case of microcellular injection molding—the TPEC of the 20 and 30 wt% composites increased by 7–74 and 18–32 times, respectively; (3) although microcellular injection molding alone (i.e., without GCP) showed the greatest influence on the randomness of the fiber orientation and the TPEC, the TS of the samples was the lowest due to the uncontrollable foaming cell size and cell size uniformity; (4) in contrast, when GCP was employed in the microcellular foaming process, high TS was obtained, and the TPEC was significantly enhanced. The high foaming quality owing to the GCP implementation improved the randomness of fiber orientation, as well as the electrical and mechanical properties of the composites. Generally speaking, microcellular injection combined with gas counter pressure does provide a promising way to achieve high electrical and mechanical performance for carbon-fiber-added polypropylene composites.

## 1. Introduction

Conductive polymer composites are promising materials for many applications, such as rechargeable batteries, electromagnetic interference shielding, electrically conductive or static dissipative sensors, and electronic packaging devices [1,2,3,4]. Conductive polymer composites can be obtained by blending conductive additives, such as carbon fiber (CF) [5,6], carbon black [7,8], stainless steel fibers [9,10], and/or carbon nanofibers and nanotubes [11,12] with a polymer matrix. Although nanoscale additives are of great interest in polymer composites, owing to their superior electrical properties, the high processing costs limit their applications [13]. To date, microsized additives, particularly CF, are the most cost effective and widely used conductive additives [5,6].

The conductivity of composites reinforced with discontinuous fibers can be varied over a wide range (from insulating to highly conducting), depending on various factors [5,6,7,8,9,10,11,12,13,14,15,16,17,18,19,20,21,22,23,24]. Gurland [14] reported that the electrical conductivity of conductive polymers increases linearly with the fiber concentration until a critical concentration, above which a nonlinear relationship is observed [15]. This critical concentration depends mainly on the electrical characteristics of the filler. Davenport [17] also reported that conductivity increases with an increase in the fiber aspect ratio because a conductive path can be generated more easily at higher aspect ratios. Heinzel et al. [20] and Blunk et al. [21] reported that the electrical conductivity of conductive polymers varies with the fiber orientation. Fiber orientations in an injection-molded part mainly depend on the flow field, including the part geometry and injection speed during the injection mold filling process. The melt and mold temperatures are only secondary factors in conductive polymers [22]. Recently, Chen et al. proposed gas counter pressure (GCP) technology [23,25], which influences the fiber orientation and the associated through-plane electrical conductivity (TEPC) of polymers. GCP restricts fountain flows around the melt front and thus results in a more random fiber orientation [23] and higher TEPC. Ameli et al. [6] also reported that foaming can significantly increase TEPC, depending on the cell density. Recently, Yilmaz et al. [26] reported the microcellular foaming of polycarbonate composites filled with glass fiber and carbon black. Most studies have focused on the tensile properties of composites formed by micromolecular foaming. However, the effects of foaming cells on fiber orientation and the associated conductivities have not been reported. In this study, we investigated the effects of foaming in microcellular (MuCell) injection molding on the fiber orientation and the associated TEPC and tensile performance of molded polypropylene (PP) samples with various contents of CF additives. CIM was also employed, and selected results were compared. Then, GCP combined with MuCell injection molding was also conducted to investigate its influence on the properties of the molded parts. In general, more random fiber orientation results in less tensile strength (TS) along the melt flow direction. Therefore, the effects of GCP, MuCell injection molding, and MuCell injection molding plus GCP on the TS of the molded parts along the flow direction were also examined.

## 2. Experimental Section

The experiment was conducted following procedures reported in previous studies [16,23,24,25,27].

### 2.1. MuCell Injection Molding Machine and Gas Counter Pressure Regulation

An ARBURG ALLROUNDER 420C (ARBURG GmbH + Co KG, Arthur-Hehl-Straße, 72290 Lossburg, Germany) (Figure 1a) equipped with a supercritical fluid (SCF) generator (Trexel; Figure 1b) was used. The machine uses nitrogen as a foaming agent. A homemade gas pressure regulating unit with a high-frequency gas control valve was used to provide the required GCP (Figure 1c).

### 2.2. Experimental Mold

A bar-shaped mold was used to fabricate specimens for tensile tests (ASTM-D638 Type I), as shown in Figure 2. A single fan gate was designed at the upper side of the mold, and an O-ring (orange line in Figure 2a) was used to seal the parting surface when applying GCP. A GCP air inlet was designed at the lower side of the mold (Figure 2c), with an overflow area in the front. The molded specimen (Figure 2b) was cut at the center to analyze the fiber orientation and the through-plane electrical conductivity (TPEC).

### 2.3. Materials

In this study, polypropylene was offered from Taiwan Chemical with the product name K-1035. Carbon fiber was manufactured by the Japanese company Toho Tenax Co. Ltd. (Teijin Limited, Kasumigaseki Common Gate West Tower, 3-2-1 Kasumigaseki, Chiyoda-ku, 100-8585 Tokyo, Japan) with the brand name TENAX@C493. Samples with 20 and 30 wt% of the filler were compounded by YE-HO-JI Plastic, a professional compounding company, using the standard twin-screw extruder compounding method.

### 2.4. Characterization

A universal tensile testing machine (MTS Criterion Model 43) (MTS Systems Corporation, 14000 Technology Drive, Eden Prairie, MN 55344, USA) was used in this study to measure the TS of the specimens. The tensile speed was 0.0625 mm/s, and other settings follow the ASTM D638 Type I standard.

The through-plane resistance (TPR) of the specimen was measured using the contact resistance method; the sample was measured from the front plant to the back plant and into an insulated fixture electrified at 1 amp. The TPR was calculated using Ohm’s law, and the TPEC was calculated as the reciprocal of the TPR.

The fiber orientation was analyzed following the procedure reported in our previous study [23]. The fiber orientation level (FOL) was used as an index to evaluate the correlation of the TPEC with the fiber orientation. A scanning electron microscope (SEM; Hitachi, Model S-3000N, 15 KV) (Hitachi High-Tech Corporation, Toranomon Hills Business Tower, 1-17-1 Toranomon, Minato-ku, Tokyo 105-6409, Japan) and a gold-foil-coating unit (E-1010) were used to characterize the fiber orientation and foaming cell at the cross section of the specimens. Magnified SEM images (150×) of the specimens were also obtained.

SEM images of polished surfaces of the specimens were quantitatively analyzed using Image. All the samples were coated with gold before the SEM analysis. Figure 3 shows the SEM image used to examine the fiber orientation (half gap across the thickness section) of the specimens. To evaluate the fiber orientation degree, six orientation angle ranges (0–30°, 31–60°, 61–90°, 91–120°, 121–150°, and 151–180°) were analyzed [23]. The 0° and 90° angles indicate CFs parallel and perpendicular to the flow direction, respectively. Considering the images from the LW (length–width) plane, the FOL was calculated using Equation (1) [5,23]:(1)$$\mathrm{FOL}=\frac{\sum_{1}^{n}{\left(\cos\theta \right)}_{n}}{n}$$
where θ is the fiber in-plane angle with respect to the length of the specimen and was directly determined using the image analyzer, and n is the number of measured fibers. FOLs of 0 and 1 represent CFs perpendicular and parallel to the length of the specimen, respectively. Details of the calculation are shown in a previous report [23].

### 2.5. Experimental Parameters

First, we evaluated the effects of the molding parameters (melt temperature, mold temperature, and injection speed) on the fiber orientation and the TPEC of the PP composites, with 20 and 30 wt% CF additives molded by CIM. The parameters are listed in Table 1 (ID 1–7). GCP was also applied to the molding process to investigate its effects. The corresponding process parameters are listed in Table 1 (ID 8–12). Next, MuCell injection molding was employed to mold the composites under the processing conditions listed in Table 2 (ID 13–21). GCP was applied at different gas holding times to investigate their effects on the properties of the specimens (Figure 2c, ID 22–26).

## 3. Results and Discussions

### 3.1. Conventional Injection Molding

Figure 4a–f shows the SEM images of the PP/CF composites with 20 and 30 wt% fiber, respectively, fabricated at various injection speeds. The figures show the fiber orientations. The fiber was more oriented along the flow direction as the injection speed increased. The corresponding TPECs and TSs are depicted in Figure 5a,b, respectively. The higher the degree of fiber orientation (i.e., the higher the FOL), the lower the TPEC and the higher the TS.

High mold temperatures resulted in thinner skin layers and less shear stress in the core region (i.e., less orientation along the flow direction); thus, the TPEC slightly increased. A similar result was obtained at high melt temperatures. High melt temperatures resulted in low melt viscosity and shear stress reduction, and the TPEC decreased. Due to the limitation of the article length, the variation in FOL with various processing parameters and the effects of the melt and mold temperatures are summarized in Table 3.

### 3.2. Microcellular Injection Molding

CF orientations in the samples obtained via MuCell injection molding at different injection speeds are shown in the SEM images in Figure 6a–f. Cell foaming induced a pushing force on the fiber, resulting in a more random distribution (schematic is shown in Figure 6g,h) and high TPEC, as shown in Figure 5a (also in Figure 7a but on a different scale). Microcellular foaming does show a significant effect on the fiber orientation, and the TPEC was significantly improved (Figure 5a). In contrast, the TS significantly decreased (Table 3, ID 13–21). This is attributed to the uncontrolled foamed cell, and its effect exceeded the fiber orientation.

At higher SCF dosages, the foaming cell density and size increased, and the corresponding fiber orientation randomness and TPEC increased (Figure 7b). However, with very high SCF dosages, the formation of larger foaming cells decreased the effects of the dosage. Figure 8a–f shows the SEM images of the samples. For the MuCell injection molding process, the injection speed, melt temperature, and mold temperature exhibited similar effects as those of the CIM. However, they also affected the foaming characteristic, and the effects are not as straightforward as those of the CIM. In general, microcellular foaming enhanced the TPEC by 5–8 and 18–78 times for the 30 and 20 wt% CF composites, respectively.

### 3.3. Effect of Gas Counter Pressure on Conventional Injection Molding

Employing GCP alone with CIM could affect the fiber orientation and the TPEC. Compared with CIM, the TPECs of the samples with 30 and 20 wt% increased by 2–3 times, respectively. GCP slightly decreased the TS of the samples (Table 3, ID 8–12) due to the reduction of flow-direction-orientated fibers.

### 3.4. Effect of Gas Counter Pressure on Microcellular Injection Molding

Figure 9a–h shows SEM images of MuCell-molded samples under different GCP parameters, showing the foaming and fiber orientations. The combined effects of GCP and gas holding time on the associated TPEC and TS are shown in Figure 10a,b, respectively. The combined effects of GCP and gas holding time influenced the TPEC and TS more significantly than the effects of CIM and CIM plus GCP. In the sample with 20 wt% CF, the combined effects did not enhance TEPC as significantly as that obtained using MuCell injection molding alone (Table 4); however, the TPEC of the sample with 30 wt% CF was significantly enhanced (Table 4). Although microcellular foaming alone could randomize the fiber orientation, by increasing the TPEC, the associated TS also decreased. By combining GCP with MuCell injection molding, the fine and uniform foaming cells enhanced the TPEC while maintaining the TS of the samples (Table 5).

## 4. Conclusions

In this study, we prepared PP/CF (20 and 30 wt%) composites using CIM and MuCell injection foaming under various molding conditions. We investigated the TPEC and TS of the composites while considering the fiber orientations. GCP was also employed in both CIM and MuCell injection molding processes to investigate its influence on the fiber orientation, foaming cell quality, and the associated TPEC and TS.

### 4.1. Influence from Conventional Injection Molding with and without Gas Counter Pressure

In CIM, the injection speed influenced the fiber orientation and the TPEC more than the mold and melt temperatures. Increasing injection speed results in higher shear stress and a higher percentage of fiber orientation in the flow direction and ends up with lower TPEC and higher TS. Higher mold and melt temperatures reduced shear stress due to the thinner skin layer and lower viscosity, thereby increasing TPEC slightly. GCP also influenced the fiber orientation. The TEPCs of the samples with 20 and 30 wt% of CF increased by 2–3 times. However, GCP reduces the TS values slightly due to the reduction of flow-orientated fibers.

### 4.2. Influence from Typical Microcellular Injection Molding

When microcellular foaming was employed, the foaming cell significantly influenced the fiber orientation, and the TEPCs of the samples containing 20 and 30 wt% CF were increased by 18–78 and 5–8 times, respectively. High injection speeds during the MuCell injection molding process, on the other hand, decreased TEPC, similar to the case of CIM. The mold and melt temperatures showed similar trends to those of CIM. The SCF dosage showed a varying effect on the TPEC: at high SCF dosages, TPEC increased, but when the SCF dosage was too high, the big, foaming cell reduced its influence on the TPEC.

### 4.3. Influence from Combined Gas Counter Pressure with Microcellular Injection

Although GCP in the MuCell injection molding process showed less significant effects than microcellular foaming, it does improve the TS of MuCell-molded samples more significantly than that of the CIM-molded ones. By employing GCP in the microcellular foaming process, the TS of the samples increased, and the TPECs of the samples with 20 and 30 wt% increased by 7–74 and 18–32 times, respectively. This can be attributed to the foaming cell quality induced by GCP.

## Figures and Tables

**Figure 1 polymers-14-03251-f001:**
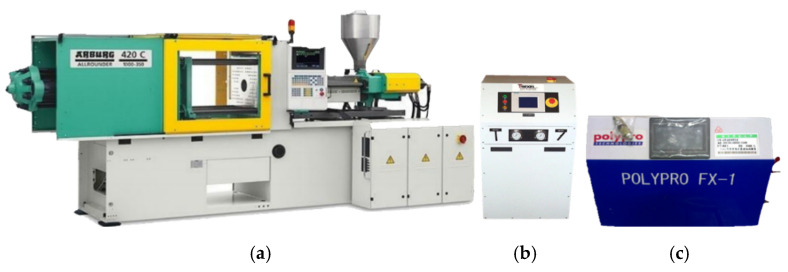
(**a**) Microcellular (MuCell) injection molding machine with a gas counter pressure equipment (ARBURG ALLROUNDER 420C). (**b**) Supercritical fluid (SCF) generator (Trexel). (**c**) High-frequency gas control valve.

**Figure 2 polymers-14-03251-f002:**
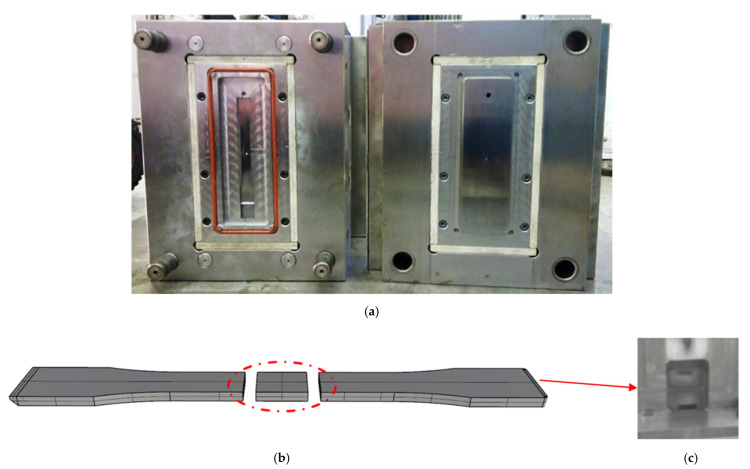
(**a**) Mold for the tensile test samples. (**b**) Specimen for fiber orientation and through-plane electrical conductivity (TPEC) analyses. The specimen is 143 mm × 19 mm × 3 mm, following the ASTM-D638 Type I standard. (**c**) Gas counter pressure (GCP) inlet position.

**Figure 3 polymers-14-03251-f003:**
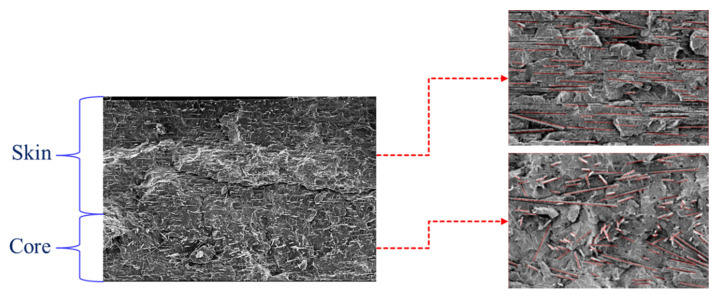
Scanning electron microscopy (SEM) image for evaluating the fiber orientation level (FOL) of the sample. The magnification is ×100.

**Figure 4 polymers-14-03251-f004:**
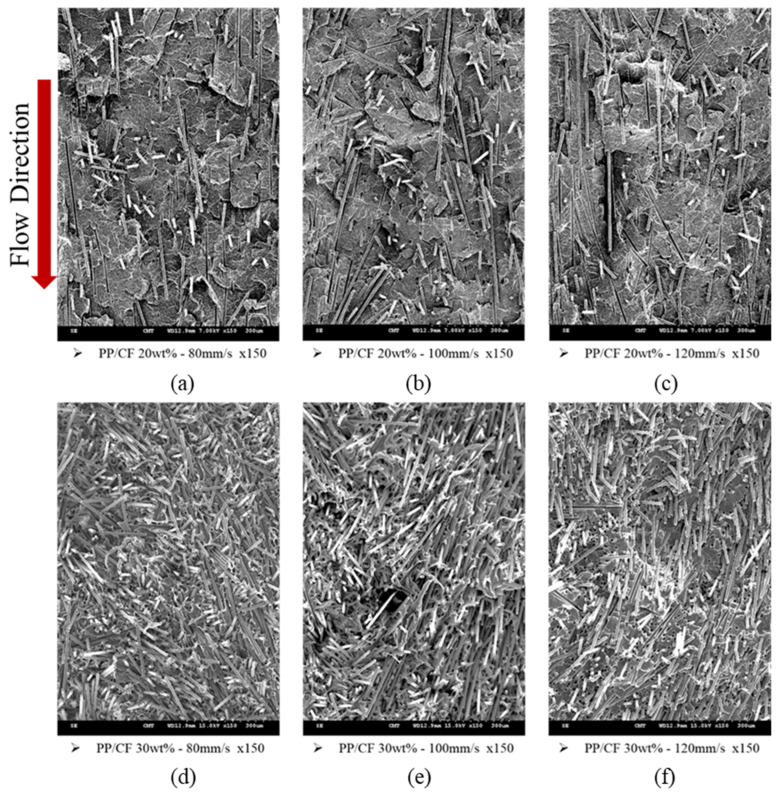
Fiber orientations under various injection speeds. (**a**) Polypropylene (PP)/carbon fiber (CF) 20 wt%, 80 mm/s. (**b**) PP/CF 20 wt%, 100 mm/s. (**c**) PP/CF 20 wt%, 120 mm/s. (**d**) PP/CF 30 wt%, 80 mm/s. (**e**) PP/CF 30 wt%, 100 mm/s. (**f**) PP/CF 30 wt%, 120 mm/s.

**Figure 5 polymers-14-03251-f005:**
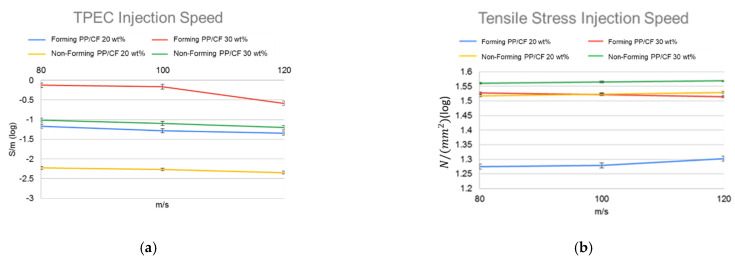
(**a**) TPEC and (**b**) tensile strength of samples with different CF contents at different injection speeds (for comparison, TPEC are presented in log scale). Comparisons for MuCell-molded (foaming) parts and CIM-molded parts (non-foaming).

**Figure 6 polymers-14-03251-f006:**
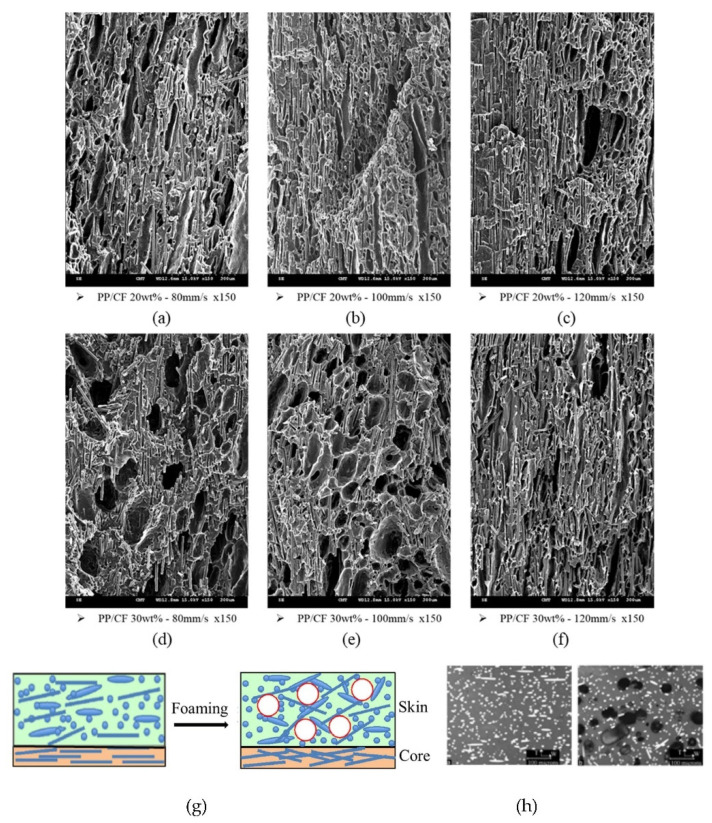
Fiber orientations with SCF-foamed PP/CF composites at various injection speeds. (**a**) PP/CF 20 wt%, 80 mm/s. (**b**) PP/CF 20 wt%, 100 mm/s. (**c**) PP/CF 20 wt%, 120 mm/s. (**d**) PP/CF 30 wt%, 80 mm/s. (**e**) PP/CF 30 wt%, 100 mm/s. (**f**) PP/CF 30 wt%, 120 mm/s. (**g**) Influence of cell foaming on fiber orientation (Schematic). (**h**) Real case.

**Figure 7 polymers-14-03251-f007:**
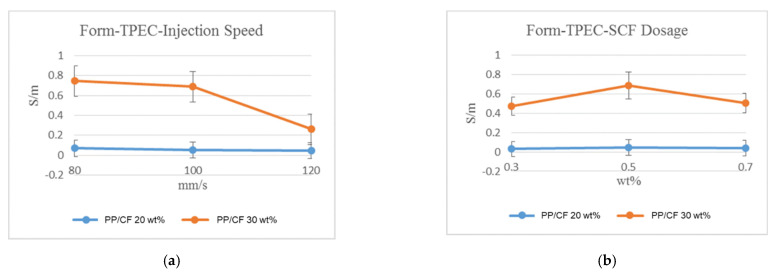
Values of TPEC after SCF foaming under various injection speeds, SCF dosage, and CF content. (**a**) Different injection speeds and (**b**) different SCF dosage.

**Figure 8 polymers-14-03251-f008:**
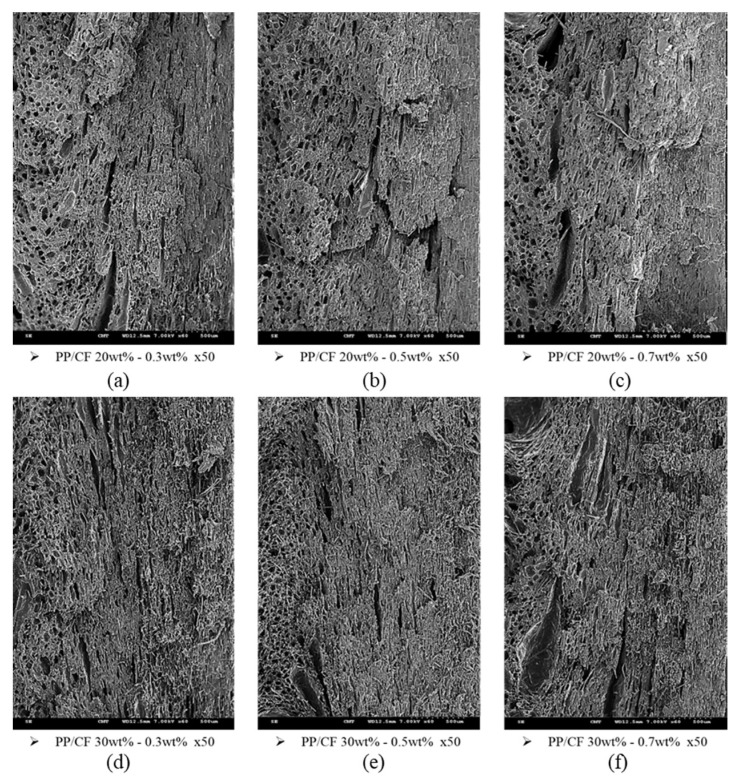
Fiber orientations in SCF-foamed PP composites with different SCF dosages: (**a**) PP/CF 20 wt%, SF 0.3 wt%; (**b**) PP/CF 20 wt%, SF 0.5 wt%; (**c**) PP/CF 20 wt%, SF0.7 wt%; (**d**) PP/CF 30 wt%, SF 0.3 wt%; (**e**) PP/CF 30 wt%, SF 0.5 wt%; (**f**) PP/CF 30 wt%, SF 0.7 wt%.

**Figure 9 polymers-14-03251-f009:**
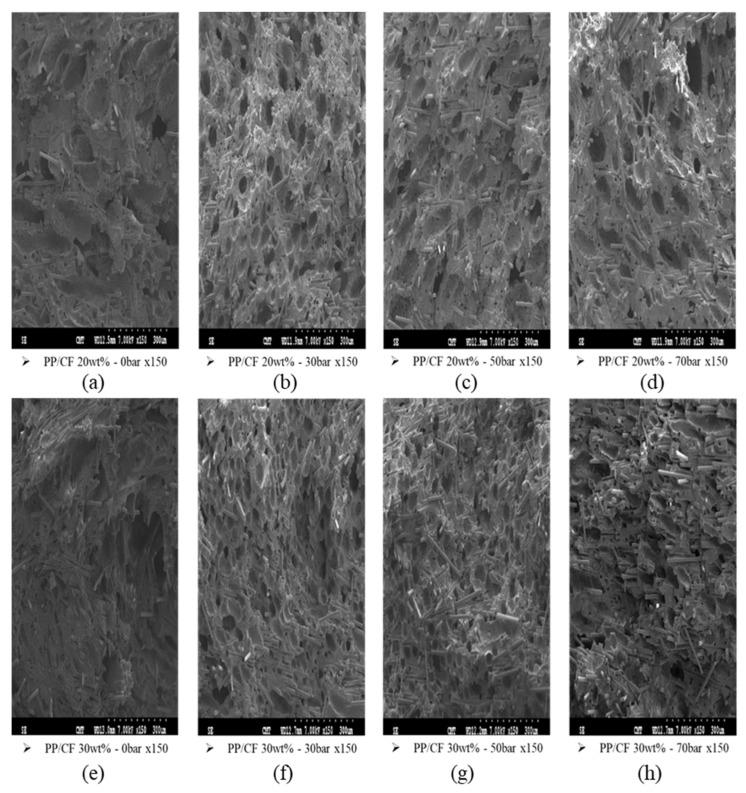
Fiber orientations in SCF-foamed PP composite with different GCPs: (**a**) PP/CF 20 wt%, GCP 0 bar; (**b**) PP/CF 20 wt%, GCP 30 bar; (**c**) PP/CF 20 wt%, GCP 50 bar; (**d**) PP/CF 20 wt%, GCP 0 bar; (**e**) PP/CF 30 wt%, GCP 0 bar; (**f**) PP/CF 30 wt%, GCP 30 bar; (**g**) PP/CF 30 wt%, GCP 50 bar; (**h**) PP/CF 30 wt%, GCP 70 bar.

**Figure 10 polymers-14-03251-f010:**
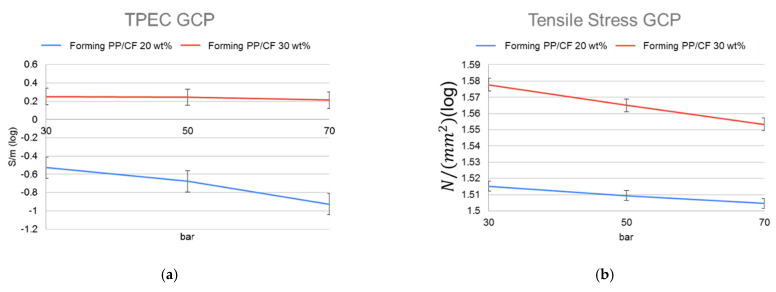
(**a**) TPEC and (**b**) TS of samples with different GCPs and CF contents.

**Table 1 polymers-14-03251-t001:** Processing conditions for conventional injection molding (CIM) without (ID 1–7) and with gas counter pressure (GCP, ID 8–12).

Group ID	Injection Speed (mm/s)	Mold Temperature (°C)	Material Temperature (°C)	GCP (bar)	GCP Holding Time (s)
1	80	50	220	0	0
2	100				
3	120				
4	100	30			
5		70			
6		50	200		
7			240		
8			220	30	3
9				50	
10				70	
11				50	1
12					5

**Table 2 polymers-14-03251-t002:** Processing conditions for MuCell injection molding without (ID 13–21) and with GCP (ID 22–26).

Group ID	Injection Speed (mm/s)	Mold Temperature (°C)	Material Temperature (°C)	SCF Dosage (wt%)	GCP (bar)	GCP Holding Time (s)
13	80	50	220	0.5	0	0
14	100					
15	120					
16	100	30				
17		70				
18		50	200			
19			240			
20			220	0.3		
21				0.7		
22				0.5	30	3
23					50	
24					70	
25					50	1
26						5

**Table 3 polymers-14-03251-t003:** FOL, TEPC, and TS for the experiments.

		FOL (Skin)%		TPEC (S/m)		Tensile Stress (N/mm^2^)	
	Group	20 wt%	30 wt%	20 wt% (×10^−3^)	30 wt% (×10^−2^)	20 wt%	30 wt%
Results of Conventional Injection Molding	1	0.8 ± 0.1	0.68 ± 0.15	5.9 ± 0.562	9.69 ± 1.9	32.96 ± 2.65	36.38 ± 2.39
	2	0.8 ± 0.23	0.72 ± 0.07	5.4 ± 0.921	8.04 ± 1.4	33.39 ± 3.28	36.75 ± 6.05
	3	0.89 ± 0.18	0.75 ± 0.14	4.5 ± ,0.382	6.27 ± 1.8	33.86 ± 8.26	37.07 ± 7.33
	4	0.84 ± 0.13	0.73 ± 0.18	4.7 ± 0.369	6.55 ± 1	28.07 ± 2.47	27.51 ± 7.89
	5	0.88 ± 0.19	0.7 ± 0.05	5.6 ± 0.125	7.64 ± 1.9	29.52 ± 1.73	28.44 ± 3.64
	6	0.85 ± 0.1	0.74 ± 0.07	4.3 ± 0.799	4.58 ± 3.7	28.05 ± 3	27.42 ± 7.85
	7	0.89 ± 0.21	0.68 ± 0.19	5.2 ± 0.661	7.57 ± 1.5	28.86 ± 5.46	27.94 ± 7.89
Results of Conventional Injection Molding Combined with GCP	8	0.63 ± 0.04	0.66 ± 0.15	15.2 ± 0.292	17.56 ± 2.3	35.32 ± 6.79	39.66 ± 5.53
	9	0.65 ± 0.06	0.69 ± 0.13	14.8, ± 0.417	15.53 ± 3.8	34.76 ± 3.6	38.45 ± 2.32
	10	0.68 ± 0.12	0.72 ± 0.19	14.6 ± 0.26	13.11 ± 3.5	34.38 ± 8.63	37.22 ± 6.53
	11	0.64 ± 0.19	0.65 ± 0.09	14.6, ± 0.269	15.86 ± 8	36.26 ± 7.37	39.57 ± 3.67
	12	0.71 ± 0.12	0.74 ± 0.1	13.5 ± 0.104	11.31 ± 2.88	35.57 ± 3.9	38.12 ± 9.61
Results of Microcellular Injection Molding	13	0.65 ± 0.11	0.65 ± 0.11	68.1 ± 0.682	74.69 ± 15.5	18.81 ± 1	18.7 ± 1.36
	14	0.68 ± 0.09	0.67 ± 0.11	51.9 ± 0.103	68.7 ± 19.9	19.02 ± 2.44	19.53 ± 2.03
	15	0.69 ± 0.16	0.68 ± 0.16	45.1 ± 0.325	26.12 ± 3.54	20.06 ± 2.63	21.2 ± 4.95
	16	0.64 ± 0.17	0.65 ± 0.08	386 ± 36.2	46.01 ± 12.15	19.51 ± 3.53	20.21 ± 3.72
	17	0.67 ± 0.09	0.68 ± 0.12	42.4 ± 4.81	54.07 ± 15.18	18.42 ± 2.81	19.29 ± 3
	18	0.63 ± 0.1	0.63 ± 0.13	35 ± 6.07	38.18 ± 6.2	23.23 ± 1.75	20.28 ± 2.53
	19	0.68 ± 0.17	0.67 ± 0.15	40.4 ± 6.98	61.14 ± 4.66	20.43 ± 2.55	19.84 ± 4.64
	20	0.69 ± 0.16	0.67 ± 0.05	31 ± 4.28	47.48 ± 13.51	20.53 ± 3.41	18.9 ± 2.76
	21	0.64 ± 0.12	0.64 ± 0.13	39.7 ± 9.69	50.86 ± 12.35	19.85 ± 4.64	18.49 ± 0.99
Results of Microcellular Injection Molding Combined with GCP	22	0.66 ± 0.15	0.69 ± 0.19	296.8 ± 82.8	178.65 ± 18.73	32.75 ± 5.98	37.82 ± 6.57
	23	0.68 ± 0.09	0.71 ± 0.1	227.5 ± 66.2	175.58 ± 33.97	32.31 ± 2.24	36.73 ± 10.91
	24	0.7 ± 0.12	0.75 ± 0.09	118.6 ± 7.15	162.46 ± 37.65	31.96 ± 1.83	35.75 ± 3.97
	25	0.66 ± 0.13	0.69 ± 0.08	216.9 ± 50.4	170.15 ± 47.06	32.52 ± 8.49	37.68 ± 5.8
	26	0.73 ± 0.14	0.76 ± 0.08	112.6 ± 29.9	161.16 ± 30.16	31.86 ± 8.64	35.46 ± 3.33

**Table 4 polymers-14-03251-t004:** Comparisons of TPECs of samples molded with different processing techniques.

	TPEC (S/m) (Scope)	Enhancement (Times)
PP/CF 20 wt% no foaming	0.0043–0.0052	1
PP/CF 20 wt% foaming	0.0803–0.4056	18.67–78.00
(GCP)PP/CF 20 wt% no foaming	0.0132–0.0152	2.923–3.069
(GCP)PP/CF 20 wt% foaming	0.0310–0.3860	7.209–74.23
PP/CF 30 wt% no foaming	0.0458–0.0969	1
PP/CF 30 wt% foaming	0.2312–0.7977	5.048–8.232
(GCP)PP/CF 30 wt% no foaming	0.1131–0.1756	1.812–2.469
(GCP)PP/CF 30 wt% foaming	1.4996–1.7856	18.43–32.74

**Table 5 polymers-14-03251-t005:** Comparisons of TS molded under various processing technology.

	$\mathrm{Tensile}\;\mathrm{Strength}\;(N/{\mathrm{mm}}^{2})$	Enhancement (times)
PP/CF 20 wt% no foaming	28.05–33.86	1
PP/CF 20 wt% foaming	18.42–23.35	0.66–0.69
(GCP)PP/CF 20 wt% no foaming	34.38–36.26	0.934–0.942
(GCP)PP/CF 20 wt% foaming	31.86–33.68	0.99–1.14
PP/CF 30 wt% no foaming	27.42–36.75	1
PP/CF 30 wt% foaming	18.49–21.20	0.58–0.67
(GCP)PP/CF 30 wt% no foaming	37.47–39.66	0.96–0.972
(GCP)PP/CF 30 wt% foaming	34.72–38.77	1.06–1.27

## Data Availability

Not applicable.

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
