# Peer review of "Processing Effects on the Through-Plane Electrical Conductivities and Tensile Strengths of Microcellular-Injection-Molded Polypropylene Composites with Carbon Fibers"

_polymers, 2022, doi:10.3390/polym14163251_

Round 1

Reviewer 1 Report

Even though the study of how processing condition affect the fiber orientation and the electrical and mechanical properties of a composite is interesting, the paper presented by the authors has a lot of major flaws and problems: this drives me to a rejection decision. Some of my comments:

There is not any standard deviation on the measuraments. As the variation of the properties is tiny (e.g. typically a relevant variation in electrical conductivity is reported as “order of magnitude” not in “times higher”), I wonder if the discussed effect is real or just covered by the experimental error.

The organization of the manuscript is really confusing and there is not any support by the authors to help understanding the results (e.g. there are many acronym which at the end I have difficulty to remember; the results are reported in Table 4 reporting the testing code, which is difficult to correlate with what every experiment corresponds to). Discussion section should be strongly revised.

I would remove Fig 3 or make just one with fig 2 (reducing the images, which are not useful in my opinion)

Tensile test speed is really low, are the authors sure about that?

English is really poor and need to be revised.

Author Response

Replication to the review comments SI-Polymer

________________________________________________________________

Even though the study of how processing condition affect the fiber orientation and the electrical and mechanical properties of a composite is interesting, the paper presented by the authors has a lot of major flaws and problems: this drives me to a rejection decision. Some of my comments:

[R] Very sorry about that. We may upload the wrong version (not final one) during the submission and make you confused.

There is not any standard deviation on the measuraments. As the variation of the properties is tiny (e.g. typically a relevant variation in electrical conductivity is reported as “order of magnitude” not in “times higher”), I wonder if the discussed effect is real or just covered by the experimental error.

[R] Sorry for that! Five molded specimens were used for each measurement, the standard deviations were shown in the figures with the error bar and the relevant values were also listed in Table 3. The variation ratio ranges from about 80 to 0.58, using one or two order of magnitude will be too rough to describe the change. This is why “times” was used.

The organization of the manuscript is really confusing and there is not any support by the authors to help understanding the results (e.g. there are many acronym which at the end I have difficulty to remember; the results are reported in Table 4 reporting the testing code, which is difficult to correlate with what every experiment corresponds to). Discussion section should be strongly revised.

[R] Table 4 has been noted with four different processing technologies. It is more worth to look at the comparisons (Table 5 and Table 6), variation due to different processing technologies. Abstract and conclusions were also highlighted. We put the subtitle in the result and discussion, that can help the reading. Moreover, a schematic regarding how foaming cell affects fiber orientation was added in Figures 6g and 6h. This can more clearly explain the mechanism. Using of acronym is minimized in the text, however, we still need it in the table because of the space issue.

I would remove Fig 3 or make just one with fig 2 (reducing the images, which are not useful in my opinion)

[R]Ok, thanks for the suggestion! We made the change.

Tensile test speed is really low, are the authors sure about that?

[R] Following the ASTM D638 Type I standard, the tensile speed was 0.0625 mm/s instead of 0.0625mm/min. Sorry for the mistake and confusing.

Reviewer 2 Report

The paper seeks to introduce an approach ‘’ Processing Effects on Through-Plane Electrical Conductivities and Tensile Strengths of Microcelluar Injection Molded Polypropylene Composites with Carbon Fibers. However, the authors should consider to improve upon the quality to further highlight and emphasis. 

1.    The title of your manuscript is too long and contains a spelling mistake. For instance, instead of “microcellular”, you wrote it as “microcelluar”.

2.    Introduce one or two lines highlighting the problem you are trying to solve at the beginning of the abstract.

3.    Based on the understanding of what should be included in the abstract, consider adding one or two lines highlighting the significance of the study.

4.    The introduction needs to be improved by relating to the mechanics of the studied materials and their mechanical characteristics. The references to be included are: 10.1007/s10853-022-06994-3, 10.1177/0021998318790093, 10.1016/j.polymertesting.2017.09.009, 10.1016/j.compstruct.2021.114698 and 10.1002/app.46770.

5.    Put space between each value and its corresponding units. Consider spacing between the values and their percentage units. From your abstract instead of “20wt%”, write it as “20 wt. %”

6.    Indicate clearly the various conditions used for the whole experiments in the abstract to summarize the whole study.

7.    In the experimental section, second paragraph, it is written as “The used injection machine was the ARBURG ALLROUNDER 420C”. Consider writing it as “the injection machine used is ARBURG ALLROUNDER 420C” and give space between pressure and the parenthesis which contains fig. 1c

8.    One standard of spelling should be adopted. You used ‘figure’ and ‘fig.’ in the manuscript. Consider adopting one style (in full or abbreviated).

9.    Under property characterization, paragraph 5, consider writing “Taking the images taken from the LW (length-width) plane” as “considering the images from the LW (length-width) plane”.

10.What is the magnification used in measuring the fiber orientation level figure 4?

 Put a hyphen between through and plane in line 4 of the conclusion

Author Response

The paper seeks to introduce an approach ‘’ Processing Effects on Through-Plane Electrical Conductivities and Tensile Strengths of Microcelluar Injection Molded Polypropylene Composites with Carbon Fibers. However, the authors should consider to improve upon the quality to further highlight and emphasis.

  1. The title of your manuscript is too long and contains a spelling mistake. For instance, instead of “microcellular”, you wrote it as “microcelluar”.

[R]Sorry about this mistake, it has been corrected.

  1. Introduce one or two lines highlighting the problem you are trying to solve at the beginning of the abstract.

[R]OK. Corrections were shown in red ink.

  1. Based on the understanding of what should be included in the abstract, consider adding one or two lines highlighting the significance of the study.

[R]OK. Corrections were shown in red ink.

  1. The introduction needs to be improved by relating to the mechanics of the studied materials and their mechanical characteristics. The references to be included are: 10.1007/s10853-022-06994-3, 10.1177/0021998318790093, 10.1016/j.polymertesting.2017.09.009, 10.1016/j.compstruct.2021.114698 and 10.1002/app.46770.

[R]Thanks for suggestion! After check with the content, only one is PP base composites (title: Experimental and modeling analysis of mechanical-electrical behaviors of polypropylene composites filled with graphite and MWCNT fillers), we will make a generic cite (ref.24)

  1. Put space between each value and its corresponding units. Consider spacing between the values and their percentage units. From your abstract instead of “20wt%”, write it as “20 wt. %”

[R]OK, thanks for the suggestion and corrections were made.

  1. Indicate clearly the various conditions used for the whole experiments in the abstract to summarize the whole study.

[R] Results were addressed in four aspects based on different processing technologies.

  1. In the experimental section, second paragraph, it is written as “The used injection machine was the ARBURG ALLROUNDER 420C”. Consider writing it as “the injection machine used is ARBURG ALLROUNDER 420C” and give space between pressure and the parenthesis which contains fig. 1c

[R] Ok, thanks!

  1. One standard of spelling should be adopted. You used ‘figure’ and ‘fig.’ in the manuscript. Consider adopting one style (in full or abbreviated).

[R] Ok, we changed to use the full name.

  1. Under property characterization, paragraph 5, consider writing “Taking the images taken from the LW (length-width) plane” as “considering the images from the LW (length-width) plane”.

[R]OK! Suggestion was done.

10.What is the magnification used in measuring the fiber orientation level figure 4?

[R] the magnification is 100 (already indicated in the Figure 3)

 Put a hyphen between through and plane in line 4 of the conclusion

[R] yes

Round 2

Reviewer 1 Report

The authors took into account most of the suggestions.

Reviewer 2 Report

The authors attempted to resolve most of the issues. However, the introduction needs to contain the advised relevant references which are:

10.1177/0021998318790093 (Tensile behaviour of composites)

10.1016/j.compstruct.2021.114698 (Recent Review on polymeric composites)

10.3390/polym14132662 (Tensile behaviour of composites)

10.1016/j.porgcoat.2022.107015 (Conductivity of composites)

Kindly make the needful edits and resubmit.

Best Regards.